# Seroprevalence of anti-Lassa Virus IgG antibodies in three districts of Sierra Leone: A cross-sectional, population-based study

Donald S. Grant[1,2☯], Emily J. Engel [3☯]*, Nicole Roberts Yerkes[3], Lansana Kanneh[1], James Koninga[1], Michael A. Gbakie[1], Foday Alhasan[1], Franklyn B. Kanneh[1], Ibrahim Mustapha Kanneh[1], Fatima K. Kamara[1], Mambu Momoh[1,4], Mohamed S. Yillah[1], Momoh Foday[1], Adaora Okoli[5], Ashley Zeoli[3], Caroline Weldon[5], Christopher M. Bishop[6], Crystal Zheng[7], Jessica Hartnett[3], Karissa Chao[6], Kayla Shore[5], Lilia I. Melnik[6], Mallory Mucci[5], Nell G. Bond[6], Philip Doyle[6], Rachael Yenni[6], Rachel Podgorski[5], Samuel C. Ficenec[8], Lina Moses[5], Jeffrey G. Shaffer[9], Robert F. Garry[6], John S. Schieffelin[3]

1 Lassa Fever Program, Kenema Government Hospital, Ministry of Health and Sanitation, Kenema, Sierra Leone, 2 College of Medicine and Allied Health Sciences, University of Sierra Leone, Freetown, Sierra Leone, 3 Department of Pediatrics, Sections of Pediatric Infectious Diseases, Tulane University School of Medicine, New Orleans, Louisiana, United States of America, 4 Eastern Technical University of Sierra Leone, Kenema, Sierra Leone, 5 Department of Tropical Medicine, Tulane University School of Public Health and Tropical Medicine, Tulane University, New Orleans, Louisiana, United States of America, 6 Department of Microbiology and Immunology, Tulane University School of Medicine, New Orleans, Louisiana, United States of America, 7 Department of Internal Medicine, Section of Infectious Diseases, Tulane University School of Medicine, New Orleans, Louisiana, United States of America, 8 Department of Internal Medicine, Tulane University School of Medicine, New Orleans, Louisiana, United States of America, 9 Department of Biostatistics and Data Science, Tulane University School of Public Health and Tropical Medicine, New Orleans, Louisiana, United States of America

☯ These authors contributed equally to this work.
* eengel@tulane.edu

**Data Availability Statement:** The data, along with a data dictionary explain each variable, is available

## Abstract

### Background

Lassa virus (LASV), the cause of the acute viral hemorrhagic illness Lassa fever (LF), is endemic in West Africa. Infections in humans occur mainly after exposure to infected excrement or urine of the rodent-host, *Mastomys natalensis*. The prevalence of exposure to LASV in Sierra Leone is crudely estimated and largely unknown. This cross-sectional study aimed to establish a baseline point seroprevalence of IgG antibodies to LASV in three administrative districts of Sierra Leone and identify potential risk factors for seropositivity and LASV exposure.

### Methodology and principal findings

Between 2015 and 2018, over 10,642 participants from Kenema, Tonkolili, and Port Loko Districts were enrolled in this cross-sectional study. Previous LASV and LF epidemiological studies support classification of these districts as "endemic," "emerging," and "non-endemic", respectively. Dried blood spot samples were tested for LASV antibodies by ELISA to determine the seropositivity of participants, indicating previous exposure to LASV. Surveys were administered to each participant to assess demographic and environmental

at the following link: https://data.cvisb.org/dataset/lassa-epi-2023.

**Funding:** This study was funded by a USAID funded program called the Partnerships for Enhanced Engagement in Research, Health (PEER, Health) under the department of Development, Security, and Cooperation in the Policy and Global Affairs. Funding started in 2013 and was awarded to DSG. The funders had no role in study design, data collection and analysis, decision to publish, or preparation of the manuscript.

**Competing interests:** The authors have declared that no competing interests exist.

factors associated with a higher risk of exposure to LASV. Overall seroprevalence for antibodies to LASV was 16.0%. In Kenema, Port Loko, and Tonkolili Districts, seroprevalences were 20.1%, 14.1%, and 10.6%, respectively. In a multivariate analysis, individuals were more likely to be LASV seropositive if they were living in Kenema District, regardless of sex, age, or occupation. Environmental factors contributed to an increased risk of LASV exposure, including poor housing construction and proximity to bushland, forested areas, and refuse.

## Conclusions and significance

In this study we determine a baseline LASV seroprevalence in three districts which will inform future epidemiological, ecological, and clinical studies on LF and the LASV in Sierra Leone. The heterogeneity of the distribution of LASV and LF over both space, and time, can make the design of efficacy trials and intervention programs difficult. Having more studies on the prevalence of LASV and identifying potential hyper-endemic areas will greatly increase the awareness of LF and improve targeted control programs related to LASV.

### Author summary

Lassa fever (LF), an acute viral hemorrhagic fever, is a major public health threat in West Africa. Lassa virus (LASV), the cause of LF, is transmitted to humans from the infected excrement or urine of the rodent-host, *Mastomys natalensis*. The true prevalence of LASV in Sierra Leone remains unknown. Working with the LF research program at Kenema Government Hospital (KGH), this study sought to establish a baseline seroprevalence of antibodies to LASV in Sierra Leone, targeting three administrative districts. Previous studies suggest LF and the presence of LASV is more widespread in Sierra Leone than previously recognized. This study corroborated these suggestions and revealed potential demographic and environmental factors that could increase the risk of exposure to LASV. As the largest epidemiological study conducted on LASV to-date in Sierra Leone, it will help inform future public health interventions and improve epidemiological, ecological, and clinical studies on LF and LASV.

## Introduction

Lassa fever (LF) is an acute viral hemorrhagic illness caused by the Lassa virus (LASV), a mammarenavirus. [1–3] LASV is a zoonotic pathogen transmitted predominantly through direct or indirect contact with the rodent-host *Mastomys natalensis* (also known as the Natal multi-mammate mouse). [4, 5] It has been detected in other rodent species, including *Mastomys erythroleucus*, *Hylomyscus pamfi*, and *Mus baoulei*. [6–8] Infections occur after exposure to household items, foodstuffs, or water contaminated by the feces and urine of the rodent-hosts. [9, 10] Butchering, hunting, and consuming undercooked rodent-host meat has also been tied to LASV infections. [11] Person-to-person transmission is infrequently reported and seen primarily as nosocomial infections, particularly in settings lacking proper resources, equipment, and awareness. [12–14] LASV infection rates are significant across West Africa, with an estimated 100,000 to 300,000 infections and 5,000 deaths annually, which cause disruptions to social, economic and public health systems. [15–17]

After major disruptions from a decade-long civil war from 1991–2002, followed by the West African Ebola Virus Disease (EVD) Outbreak in 2014–2016, Sierra Leone's public health system continues to face significant challenges. [18] A strong public health system should include sanitation and hygiene, a functioning surveillance system, health education, risk communication and awareness, and nosocomial infection prevention and control. [19–22] To help improve these systems the Ministry of Health and Sanitation of Sierra Leone (MoHS) considers LF to be one of the nation's most important public health problems and aims to improve pathogen surveillance and control.

Although LASV is endemic in the Mano River Union (which includes Sierra Leone, Guinea, and Liberia) and Nigeria, newly developed diagnostic tools and epidemiological models indicate the endemicity may extend geographically farther than previously thought. [23–25] Most signs and symptoms of LF occur 1–3 weeks after virus exposure, are highly variable, and can include fever, facial swelling, conjunctival injection, diarrhea, abdominal pain, and vomiting. Due to the non-specific clinical signs and symptoms, LF is a difficult disease to diagnose. [26] Some predictive models estimate people living in 80% of the geographical space of Sierra Leone are at risk of transmission and periodic outbreaks of LF. [4, 27] Seroprevalence studies in Eastern Sierra Leone previously demonstrated up to 52% of the population of the Eastern Province were exposed to LASV. [15] Serological evidence of LASV in other West African countries such as Mali, Ghana, Côte d'Ivoire, and Burkina Faso suggests LASV could be endemic in regions of West Africa with similar climates to Eastern and Northern Sierra Leone. [23, 28–31]. In Sierra Leone, the highest prevalence of LASV is found in Kenema District; however, representative, population-based studies of LASV throughout the country are lacking. [15] Districts to the north and west of Kenema reported confirmed LF cases as well, indicating the true seroprevalence of LASV is greatly underestimated. [32, 33]

The objective of this study was to quantify the seroprevalence of LASV in three distinct administrative areas of Sierra Leone and identify factors correlating to prevalence of LASV acquisition. Anti-LASV nucleoprotein (NP) IgG ELISAs were used to estimate the point seroprevalence in three districts predicted to have varying levels of LASV endemicity in Sierra Leone. Kenema District, historically considered to have the highest LF incidence rate in Sierra Leone, was chosen to represent an endemic district. Tonkolili District, experiencing a steady increase in cases over the last 7–10 years, was chosen as an area of emerging LF disease. [33] Port Loko District, with few confirmed LF cases reported, represents a non-endemic district in this study.

## Methods

### Ethics statement

This study obtained ethical approval from the Sierra Leone Ethics and Scientific Review Committee (SLESRC) in Freetown, Sierra Leone as well as the Tulane University Human Research Protection Office's Internal Review Board (HRPO, IRB). Upon entering each community, permission from the village leaders was obtained, followed by community outreach and information sessions before any study enrollment. All subjects willing and able to give appropriate consent, parental consent for minors, and assent (where applicable) were enrolled in this study regardless of sex or age. Under the regulations of SLESRC, the age of assent is 12-17- these participants are required to have parental permission *and* assent. Minors under the age of 12 are required to have parental permission from their parent and/or legal guardian only. Each consent, assent, or parental permission was obtained in written form in the presence of a third-party literate witness, unaffiliated with the research staff or the participant themselves. Illiterate participants consented, assented, or gave permission for a minor participant with a

thumbprint rather than a signature. All consents, assents, parental permissions, and questionnaires were administered orally in the resident's preferred local dialect or language (Krio, Mende, Temne, and Fullah). SLESRC certified oral scripts were used to ensure proper translation of the consents, assents, parental permissions, and questionnaires. All methods were carried out in accordance with good clinical practices, relevant guidelines, and regulations.

## Study design

A cross-sectional, population-based study was conducted to improve the understanding of LASV seroprevalence and potential factors correlating to prevalence of LASV acquisition. During the proposal period, a sample size assessment was conducted for each aim of the study project (Table 1). Three districts were chosen based on the estimated Kenema Government Hospital (KGH) and MoHS prevalence of Lassa fever–Kenema District (endemic for LASV), Tonkolili District (emerging for LASV), and Port Loko District (non-endemic for LASV). [15, 34–37] After stratifying the districts by community size, villages representative of small (n = 65–375), medium (n = 376–686), and large (n = 687–1000) communities were selected from each district. Village selection followed the WHO certified Epi-cluster sampling technique, selecting villages while taking into account the proportion of the districts' population sizes reported in the 2004 Sierra Leone National Census Enumeration Areas and household listings conducted by the Lassa Fever Outreach Team (LFOT). [38] This allowed for a weighted sampling to best randomize selection of households and participants and has been used in many low- to middle-income countries. The full sampling method technique for this study was a 2-stage cluster sampling, with probability proportional to size (PPS, Table A in S1 Text). [39, 40]

## Study sites and participant selection

After permission from these randomly selected communities, a sensitization on LASV transmission and LF signs and symptoms was conducted in the community before households were chosen. Sensitizations are a common way to provide reliable information and make communities or individuals aware of a certain issue, problem, or other piece of information. Within each village, households were enumerated before selection.

After enumeration, up to 25 households, defined as people living in the same dwelling and sharing meals and beds for at least six months, were randomly selected within each community. To best randomize household selection, a geographical center of the village (traditionally called a *barri*, where community members gather) was used as the focal point. The spin of a bottle determined the direction the team would start. [41] From there, every other *n*th (where *n* was a randomly chosen integer between 1 and 5) household was selected to participate. In each household, up to 12 members were asked to participate. The LFOT continually sampled

**Table 1. Sample size assessment for Lassa virus IgG seroprevalence study in Kenema, Tonkolili, and Port Loko Districts of Sierra Leone.**

| Objective | Model/test/assumptions | Outcome | Minimum required sample size | Reference |
|---|---|---|---|---|
| Estimate for LASV IgG seroprevalence in high-risk villages by age. | Two-stage multilevel model with estimated prevalence rate and intraclass correlation conservatively estimated at 50% and 30% respectively. | Estimate of true prevalence of LASV in <5, 5–15, 15–30 and >30-year age groups to within 15 percentage points with 95% confidence. | Two-stage sample of 30 villages in the first stage with and 20 households per village in the second stage. | Snijders, T. & Bosker, R. (1999). *Multilevel analysis*. London: SAGE[40] |
| Estimate for LASV IgG seroprevalence in low-risk villages by age. | Two-stage multilevel model with estimated prevalence rate and intraclass correlation conservatively estimated at 22% and 30% respectively. | Estimate of true prevalence of LASV in <5, 5–15, 15–30, and >30-year age groups to within 15 percentage points with 95% confidence. | Two-stage sample of 30 villages in the first stage with 10 households per village in the second stage. | *ibid.* |

households until all ages and sexes were adequately represented, as determined by census and MoHS data for each district. Garmin GPSMAP 64st World Wide Navigation Systems recorded GPS coordinates for each household in decimal degrees (Garmin International, Inc., Olathe, KS).

### Primary outcomes and data sources

Surveys were administered to each participant for determining demographics and household information, including housing construction, distance to potential environmental risk factors, and presence of rodent activity. Individual demographic data collected included sex, age, and occupation. A list of all variables collected can be found in Table B in S1 Text. These were self-reported orally and recorded by a member of the LFOT using standard MoHS LF questionnaires. LFOT members trained in rodent ecology related to LF assessed different environmental and socioeconomic factors. They visually inspected the construction of the house, including roof, floor, and wall materials and physically measured its proximity to environments known to be favorable to LASV rodent-hosts. These environments included bushes (wild land or forested areas), vegetable gardens and farmland, household toilet facilities, and water sources. [25, 42–44]

All questionnaires were administered orally in the language of the participant's choosing by members of the LFOT, including Krio, Mende, Temne, and Fullah. These data were recorded on structured forms and entered into a password-protected computer database. Blood collection occurred in the subject's home. After performing a finger-stick with a lancet (1.5mm depth, 30g needle gauge BD Microtainer Contact-Activated Lancet, Becton, Dickinson and Company, Franklin Lakes, NJ), sufficient capillary blood was expressed to completely fill a single spot, which can provide up to three 3-mm punches for laboratory analysis, on Whatman 903 Protein Saver cards (Whatman Ltd., Piscataway, NJ).

### Statistical analysis

Data were expressed as frequencies and percentages or means and standard deviations. IgG LASV seroprevalence was a proportion of IgG seropositivity measured at the household level. IgG LASV seropositivity was measured at the individual level. In order to select a representative sample size for each district, the number of study participants needed was determined through a 2-stage cluster sampling (pps) design. Descriptive statistics were performed at the district, chiefdom, and village levels. Pearson's Chi-Square, and Fisher's exact tests were used to compare categorical variables. Fisher's Exact tests were used to analyze data within villages due to their small sample size. Two-sample t-tests, and analysis of variance was used to compare continuous outcomes. Housing characteristics and observations were compared by categories of seroprevalence-levels in a multivariable model controlling for district, chiefdom, village, and household levels. To account for the hierarchical structure for the sampled data, the SAS System (version 9.4, SAS Institute, Inc., Cary, NC) GLIMMIX procedure was used to calculate the odds ratios. [45, 46] These analyses included interaction terms for the primary dependent variable (seropositivity or seroprevalence) and other main effects. All data were managed and analyzed using the SAS System. The type I error threshold was set at 5%.

### Laboratory analysis and methods

LASV IgG prevalence was determined from dried blood spots (DBS) by measuring the presence of IgG antibodies to the LASV NP epitope. [47–50] The DBS samples were stored at ambient temperature at KGH before shipment to Tulane University, where they were kept in a climate-controlled lab space prior to and throughout testing. The serological assay used to

determine potential exposure to LASV was a human anti-LASV immunoglobulin G (IgG) antibody enzyme-linked immunosorbent assay (ELISA). At study inception, the known lineage of LASV circulating Sierra Leone was lineage IV. [51] The manufacturer's instructions for ReLASV ELISA for human anti-LASV nucleoprotein (NP) immunoglobulin G (IgG) antibody plates were followed, with slight modifications to account for the DBS samples (Zalgen Labs, Germantown, MD). Each sample required two 3-mm punches be eluted in 300μL (one 3-mm punch in 150μL per replicate) of sample buffer rocking at 4˚C overnight. After antibody elution, 100μL per well was used for the ELISA. Samples were run in duplicate for each participant against six dilutions of a standardized calibrator. Limits of detection and quantitation of the ELISA were determined by the manufacturer. The overall sensitivity (86.3%) and specificity (97.9%) of the assay was previously determined. [52]

A four-parameter logistic regression model was used to determine a semi-quantitative antibody concentration from the optical density (OD) values. [53] This model estimated parameters as a function of the standardized optical density and negative control values per plate run. A sample was considered positive if the average of the two OD values was greater than 2.5 standard deviations above the average negative control. For plates in which the calibrators did not converge in the four-parameter logistic model, results were considered positive if they fell 2.5 standard deviations above the average negative control values for the entire data set.

## Results

Recruitment and enrollment started in July of 2015 and ended in June of 2018. In total, 10,642 individuals were enrolled from 82 villages (Fig 1). Each village averaged 11.01 (std ± 4.60) households and 129.78 (std ± 53.43) individuals. The overall sex distribution was 49.71% female, but more women enrolled from Kenema District than Tonkolili and Port Loko Districts (Table 2). There was no significant difference in the average age of participants between the three districts. The most common self-reported occupations in individuals over 15 years old were domestic work (including stay-at-home parents) and farming, followed by trade and education. Occupations for both students and teachers were classified as *education* due to

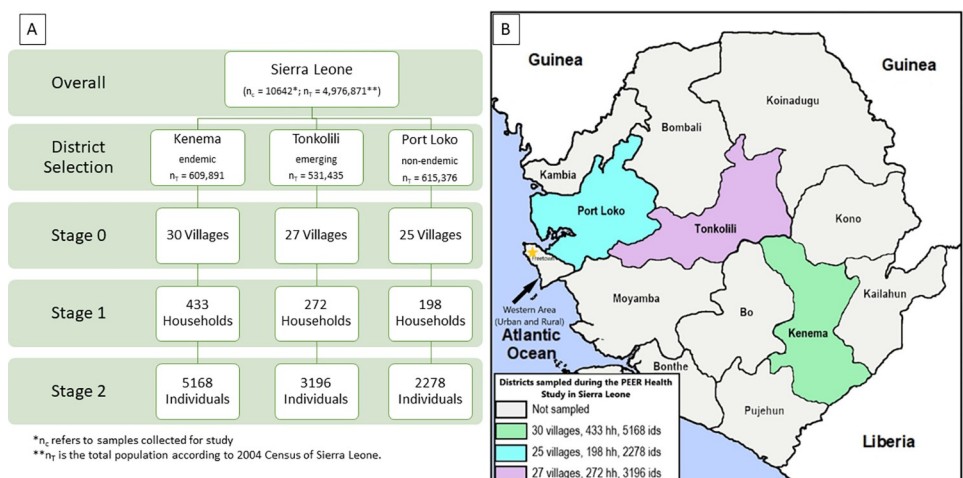

**Fig 1. Sampling method, participant selection, and district locations.** (A). Process for randomized 2-stage cluster sampling method for participant selection. (B) Map of districts visited and included in recruitment and enrollment in Sierra Leone. Base-layer map provided by Esri, OpenStreetMap contributors, HERE, Garmin, FAO, NOAA, USGS openstreetmap.org and https://cdn.arcgis.com/sharing/rest/content/items/291da5eab3a0412593b66d384379f89f/resources/styles/root.json.

**Table 2. Overall demographics of all participants, regardless of seropositivity, across Kenema, Tonkolili, and Port Loko Districts.**

| Variable | Total (n = 10642) | Kenema (n = 5168) | Tonkolili (n = 3196) | Port Loko (n = 2278) | p-value* |
|---|---|---|---|---|---|
| **Age** | | | | | |
| Mean (SE**) | 25.68 (0.20) | 25.27 (0.30) | 26.01 (0.37) | 26.17 (0.39) | .078 |
| **Sex (Female) N (%)** | 5297 (49.71) | 2675 (51.76) | 1526 (47.75) | 1089 (47.81) | < **.001** |
| **IgG Positive N (%)** | 1701 (15.98) | 1040 (20.12) | 339 (10.61) | 322 (14.14) | < **.001** |
| **Occupation *** N (%)** | | | | | |
| Farming | 2420 (35.20) | 1286 (40.59) | 632 (29.90) | 502 (31.51) | < **.001** |
| Domestic | 2047 (29.77) | 1072 (33.84) | 570 (25.96) | 405 (25.42) | |
| Trade | 1012 (14.72) | 270 (8.52) | 435 (20.58) | 307 (19.27) | |
| Education | 919 (13.37) | 255 (8.05) | 378 (17.88) | 286 (17.95) | |
| Other | 211 (3.07) | 100 (3.16) | 48 (2.27) | 63 (3.95) | |
| Mining | 169 (2.46) | 141 (4.45) | 24 (1.14) | 4 (0.25) | |
| Healthcare | 69 (1.00) | 29 (0.92) | 21 (0.99) | 19 (1.19) | |
| Transportation | 28 (0.41) | 15 (0.47) | 6 (0.28) | 7 (0.44) | |

*Chi-square and Fisher's exact test used for categorical variables. ANOVA used for comparison of mean values across districts.

**Standard error used to account for 2-stage cluster sampling, probability proportional to size.

similar environmental characteristics affecting potential LASV exposure. The trade category included stonemasonry, construction, and other vocational occupations. Overall, 1701 individuals were seropositive (16.0%) for LASV IgG (Table 2). Seroprevalence was significantly different among the three districts, with Kenema District at 20.1% seroprevalence, Tonkolili District at 10.6% and Port Loko District at 14.1% (p < .001).

Seroprevalence was visualized in ArcGIS at different levels within each district (Fig 2). When analyzed at the village level, the median seroprevalences of villages in Kenema, Port Loko, and Tonkolili Districts were 9.8% (IQR 4.7–35.2%), 9.7% (IQR 3.5–28.3%), and 6.6% (IQR 2.2–12.5%) respectively (Fig 2A). Kenema District had 10 villages (n = 30) with greater than 20% seroprevalence, which was higher than both Port Loko and Tonkolili Districts (Fig 2B). There were eight villages in Port Loko District (n = 25, Fig 2C) with greater than 20% seroprevalence, which was higher than Tonkolili District, with only four villages (n = 27, Fig 2D), though this difference was not statistically significant. In Kenema District, 42.7% (n = 433; Table C in S1 Text) of households had one or more seropositive resident, Port Loko District had 37.1% (n = 198, Table C in S1 Text) households with one or more seropositive resident, and Tonkolili District had 23.8% (n = 272, p < .001, Table C in S1 Text).

With every 10-year increase in age, participants were 1.08 (CI 1.07, 1.09) times more likely to be seropositive (p < .001, Fig 3A). This suggests about 8% of people could become IgG seropositive every 10 years. The data was then placed into age categories to help visualize this potential trend of seropositivity (Fig 4). Although a trend was noted, no significant trend was detected with a corresponding join-point regression analysis. Regardless of age, males had 1.15 (CI 1.14, 1.16) higher odds of being seropositive than females (p < .001, Fig 3A). Overall, individuals with occupations in transportation, healthcare, farming, and mining had significantly higher odds of seropositivity than domestic occupations (p-values ≤ .001, Fig 3A). Seropositivity is highly correlated with the district of residence for each participant (Fig 3A). Participants in Kenema District were 2.21 times as likely to be seropositive than individuals in Port Loko District (p < .001, Table D in S1 Text). Pairwise comparisons between districts revealed a higher proportion of seropositive individuals, regardless of age, sex, or occupation, in Kenema District (Fig 3B–3D and Tables E-G in S1 Text). Seropositive individuals were more than five

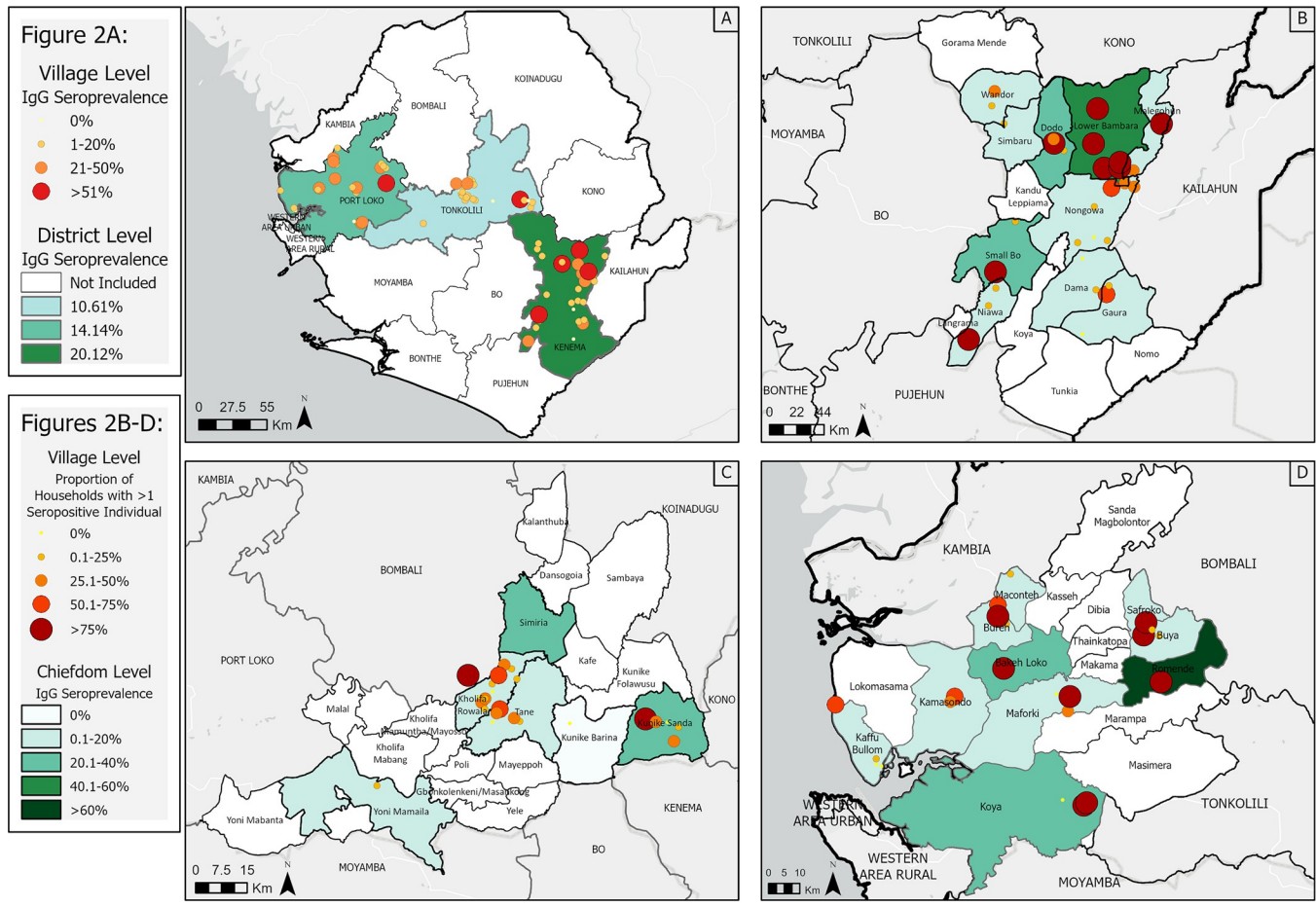

**Fig 2. Distribution of LASV IgG Seroprevalence in Sierra Leone.** (A) Mean LASV IgG seroprevalence over all three districts, stratified by village*; (B) Proportion of households with >1 LASV IgG seropositive individual by village and mean LASV IgG seroprevalence stratified by chiefdom in the endemic district (Kenema District); (C) Proportion of households with >1 LASV IgG seropositive individual by village and mean LASV IgG seroprevalence stratified by chiefdom in the emerging district (Tonkolili District); (D) Proportion of households with >1 LASV IgG seropositive individual by village and mean LASV IgG seroprevalence stratified by chiefdom in the non-endemic district (Port Loko District). Base-layer map provided by Esri, OpenStreetMap contributors, HERE, Garmin, FAO, NOAA, USGS openstreetmap.org and https://cdn.arcgis.com/sharing/rest/content/items/291da5eab3a0412593b66d384379f89f/resources/styles/root.json. *Some villages on the map are geographically close to one another. The number of villages over 20% seroprevalence can also be found in Table A in S1 Text for further reference.

times as likely to be from Kenema District if they had occupations in education, transportation, trade, or farming (Fig 3B and Table D in S1 Text).

This study also sought to better characterize potential correlates of LASV exposure. Each household was classified by their seroprevalence level, measured by the proportion of individuals in the household who tested positive for LASV IgG. The proximity of the household to environmental factors related to increased risk of exposure to LASV was compared amongst three seroprevalence-level categories–"low" (0–25% seroprevalence), "moderate" (25–50%), and "high" (>50%) (Fig 5). Households with "moderate" seroprevalence levels were more likely to be within 5-20m of the household's main water source (p = .001, Table H in S1 Text). Household proximity to refuse or garbage, bushes or wild land, vegetation or cultivated land, and household toilet facility did not have significant effects on seroprevalence levels of households. Households with "high" levels of seroprevalence were significantly more likely to source their water from a hand pump well (p < .001; Table 3) and have a rating of "Poor" for the condition of their toilet facility (p = .033, Table 3). Neither the materials used in the construction

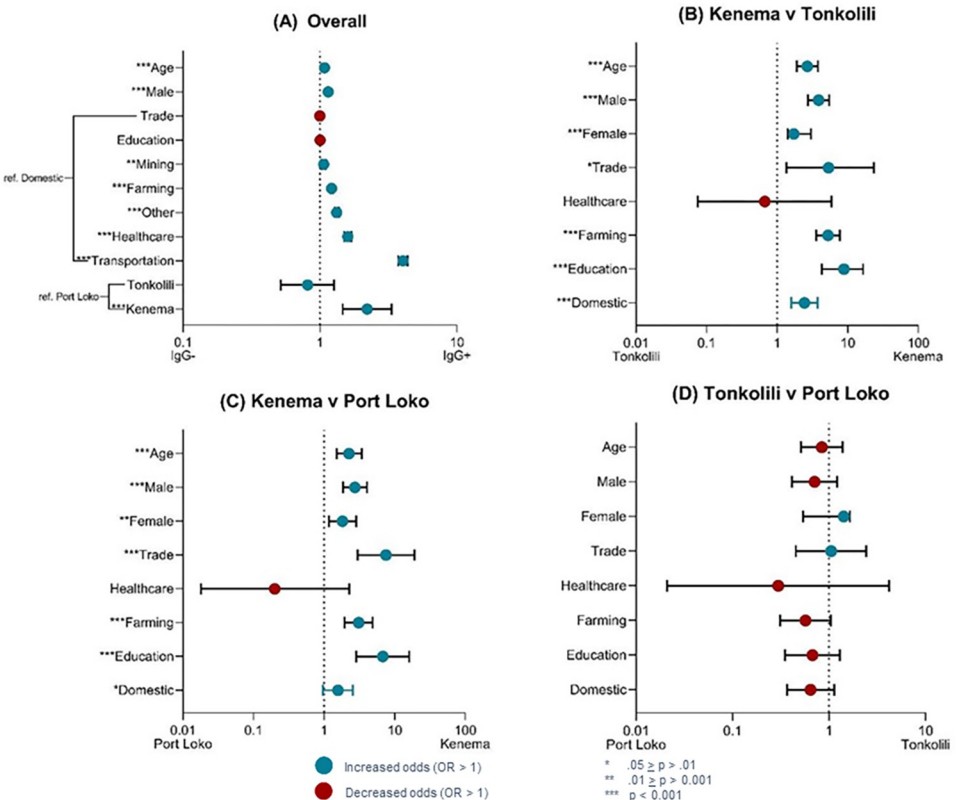

**Fig 3. Variables of interest (y-axis) and their odds ratio values (x-axis).** (A) Overall pairwise comparisons of LASV IgG seropositivity of individuals; (B) Comparing LASV IgG seropositivity of individuals in Kenema District with those in Tonkolili District; (C) Comparing LASV IgG seropositivity of individuals in Kenema District with those in Port Loko District; (D) Comparing LASV IgG seropositivity of individuals in Tonkolili District with those in Port Loko District.

of the houses, nor the fieldworker observations had significant effects on the levels of seroprevalence for the households (Table 3). When comparing individuals who live in particular households, seropositive individuals were more likely to live in houses with mud bricks (p = .013, S1A Fig), have an observation of "Poor" condition of their toilet facility (p < .001, S1B Fig), hand pump wells as their main water source (p < .001 S1C Fig), and floors made of mud only (p < .001 S1D Fig).

## Discussion

### District- and village-level seroprevalence

This cross-sectional, population-based study aimed to establish a baseline seroprevalence of LASV throughout different regions of Sierra Leone and determine whether the endemicity classifications of each district study are accurate. [34–36] LASV historically has been difficult to study due to cost, difficulty in diagnosing, and competing public health priorities. LF presents with non-specific initial symptoms often misdiagnosed as other disease, including malaria, yellow fever, and typhoid. To date, this study is the largest seroprevalence study of LASV in Sierra Leone. The ecology of LASV and its relationship to humans and development of LF is complicated, like many other endemic, zoonotic, and neglected tropical diseases. [10, 15, 24, 54, 55] Detangling these unknown dynamics, including social factors and behaviors is important; however, in order to better understand these complexities, it is imperative to

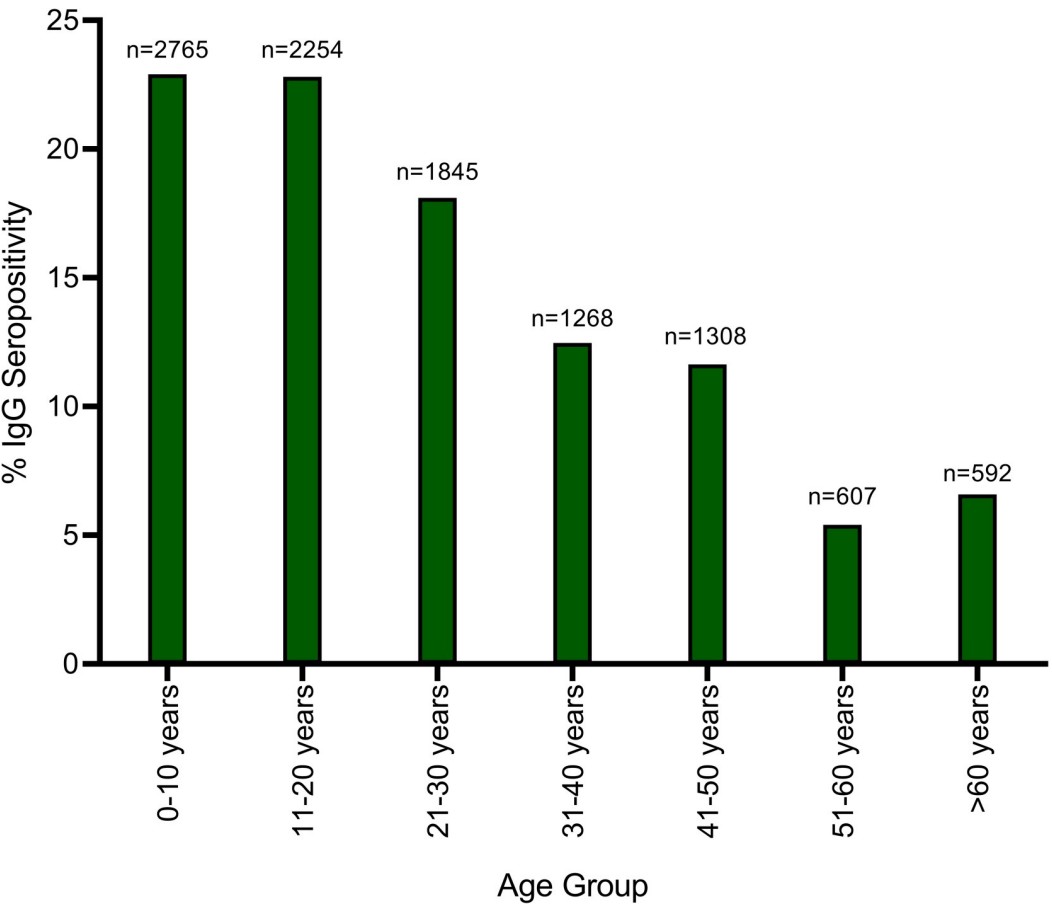

**Fig 4. LASV %IgG Seropositivity of individuals by age category.** Categories presented in 10-year age groups. A join-point regression analysis revealed no significant trend in the data. The sample size for each age group category is listed above each bar (n =).

determine which precise interactions are risk factors leading to higher rates of LASV transmission and exposure. [31, 42, 56]

Overall, the study results affirmed Kenema District is a highly endemic region for LASV, with 20.12% of individuals positive for LASV IgG. Previous studies using data from KGH Lassa Fever Diagnostic Laboratory showed a LASV IgG seroprevalence between 25.5% and 50.2% of patients presenting to the KGH VHF Isolation Ward with suspected Lassa fever cases. [57, 58] The prospective study of LASV presented in 1987 by McCormick et al. found village level seroprevalence of LASV IgG ranged from 8% to 52%. [15] In comparison, this study found a range of 0% seroprevalence to the highest at 88%. Of particular interest is the difference between the northern province in the 1987 study compared to Tonkolili District, which is part of the northern province, in this study compared to the McCormick study. McCormick et al. presented a range of 10–15% LASV seroprevalence in the north, where this study found a range contained a village with 88% seroprevalence. These differences suggest a potential change in prevalence but is not directly comparable because the methods used in the McCormick et al. study was indirect fluorescent antibody (IFA) assays rather than anti-LASV NP IgG ELISA for this study. Again, differentiating from the McCormick study, this study found multiple villages with more than 20% seroprevalence.

Each subgroup of demographics analyzed showed increased risk of seropositivity if they lived in Kenema District, compared with Tonkolili or Port Loko Districts, except individuals

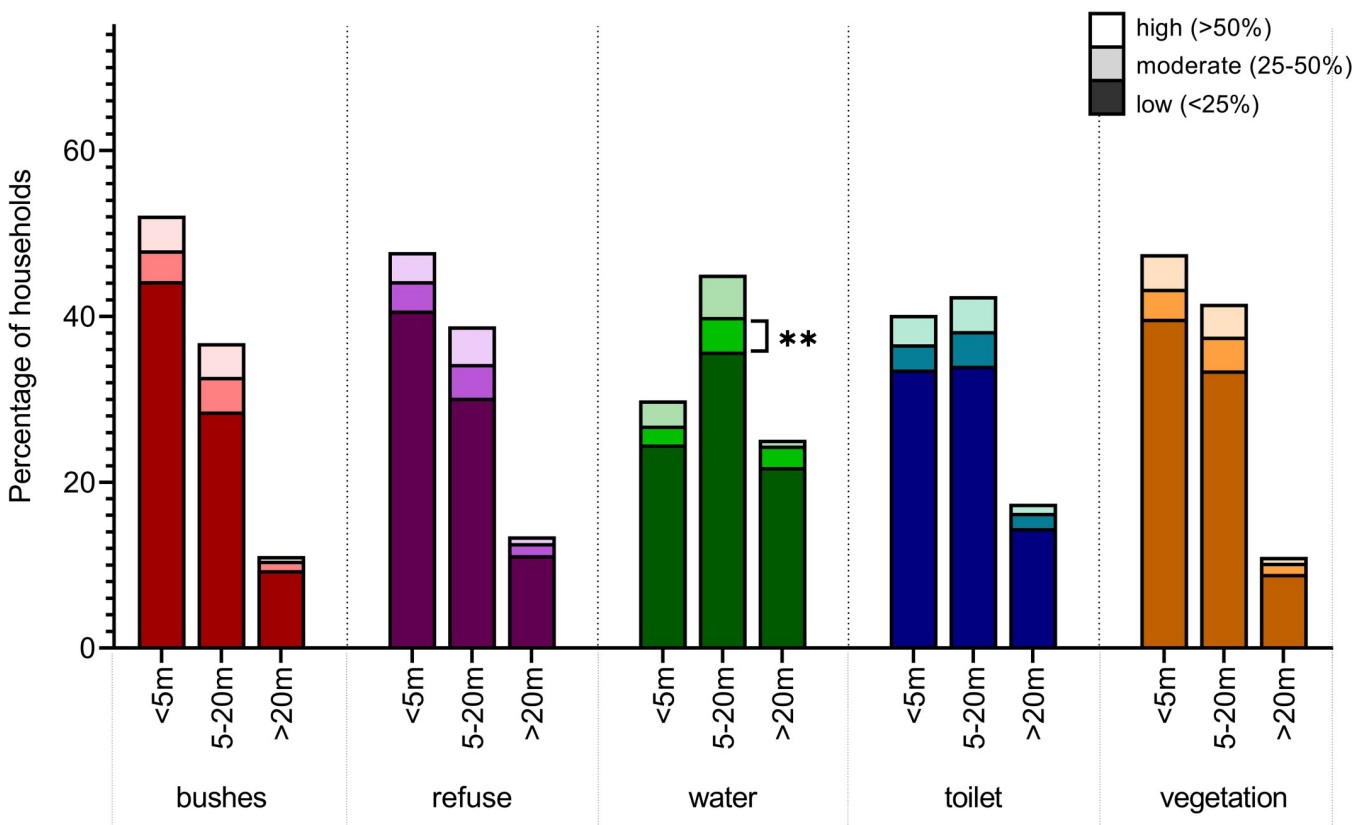

**Fig 5. Percentage of households' proximity to environmental factors, comparing household level LASV IgG seroprevalence (n = 903).** The distance of the household to the bushes or wild land, the garbage pits or refuse pile for the household, the household's main water source, the household's main toilet facility, and the household's farmland or cultivated vegetation was measured in meters by a LFOT fieldworker. Each household was categorized by the percentage of their seroprevalence. Households with more than 50% seroprevalence were categorized as high. Households with 25–50% were moderate and households less than 25% were low. The likelihood of a household having greater than 50% seroprevalence was determined in a multivariate analysis.

working in healthcare. Evidence at the chiefdom level suggests specific chiefdoms within districts could be classified as hyper-endemic due to considerably higher rates but such a classification needs further investigation. Interestingly, Port Loko District, initially chosen to represent a non-endemic district, had a higher absolute prevalence and more households with more than one sero-positive resident than Tonkolili District, considered the emerging district. Though not statistically significant, the trend shows potential for LASV to be more common in regions previously considered non-endemic. [59] LASV is known to sequester in household "hotspots," where a large proportion of LASV circulation in the rodent-host occurs in a minimal number of houses in a single community. [60] A higher proportion of households with more than one seropositive resident could indicate Port Loko District has more hotspots than Tonkolili District, increasing overall LASV seroprevalence in the district. Another reason for higher seroprevalence could be a lack of resources or understanding of LASV transmission and development of LF, but the knowledge, attitudes, and practices towards LF need further investigation. [24, 61, 62]

## Environmental correlates to LASV exposure

Increasing agricultural land use, poverty, and urbanization are known to increase the risk of exposure to LASV. [63] Along with the majority of West Africa, Sierra Leone experienced socioeconomic and demographic shifts in recent decades affecting land use, including deforestation, mining, and commercial agricultural expansion. [9, 42, 64] This implies certain

**Table 3. Percentage of households' construction materials and fieldworker observations, comparing household level LASV IgG seroprevalence (n = 903).** The likelihood of a household having greater than 50% seroprevalence was determined in a multivariate analysis. The analysis between these three groups determine potential correlates of exposure to LASV.

| Variable** | Low Seroprevalence (<25%; N(%)) | Moderate Seroprevalence (25–50%; N(%)) | High Seroprevalence (>50%; N(%)) | p-value* |
|---|---|---|---|---|
| **Water source type** | | | | |
| hand pump well | 578 (79.61) | 74 (92.50) | 52 (66.67) | < .001 |
| stream | 85 (11.71) | 6 (7.50) | 6 (7.69) | |
| tap | 47 (6.47) | 0 (0.00) | 16 (20.61) | |
| open well | 12 (1.65) | 0 (0.00) | 4 (5.13) | |
| other | 4 (0.55) | 0 (0.00) | 0 (0.00) | |
| **Toilet condition** | | | | |
| poor | 449 (63.06) | 42 (55.26) | 55 (72.37) | .033 |
| very poor | 123 (17.28) | 15 (19.74) | 10 (13.16) | |
| good | 114 (16.01) | 15 (19.74) | 5 (6.58) | |
| very good | 13 (1.83) | 1 (1.32) | 5 (6.58) | |
| fair | 13 (1.83) | 3 (3.95) | 1 (1.32) | |
| **Wall material** | | | | |
| mud bricks | 476 (65.56) | 51 (63.75) | 55 (70.51) | .458 |
| mud and stick | 224 (30.85) | 28 (35.00) | 22 (28.21) | |
| cement bricks | 26 (3.58) | 1 (1.25) | 1 (1.28) | |
| **Roof type** | | | | |
| corrugated sheets | 135 (18.60) | 11 (13.75) | 18 (23.08) | .314 |
| thatched | 591 (81.40) | 69 (86.25) | 60 (76.92) | |
| **Floor material** | | | | |
| mud only | 559 (77.00) | 61 (76.25) | 65 (83.33) | .707 |
| mud with cement | 152 (20.94) | 19 (23.75) | 12 (15.38) | |
| cement without tile | 13 (1.79) | 0 (0.00) | 1 (1.28) | |
| cement with tile | 1 (0.14) | 0 (0.00) | 0 (0.00) | |
| other | 1 (0.14) | 0 (0.00) | 0 (0.00) | |
| **Fieldworker observations*** | | | | |
| rodent holes present | 179 (24.66) | 23 (28.75) | 18 (23.08) | .678 |
| food stored indoors | 695 (95.73) | 77 (96.25) | 76 (97.44) | .736 |
| water stored indoors | 720 (99.17) | 80 (100.00) | 78 (100.00) | .306 |
| water storage covered | 298 (41.10) | 33 (41.25) | 33 (42.31) | .980 |
| rodent feces present indoors | 145 (19.97) | 13 (16.25) | 8 (10.26) | .070 |
| cement in house construction | 188 (25.93) | 19 (23.75) | 16 (20.51) | .537 |

*Presenting "yes" responses only

**Chi-square and Fisher's exact test used for categorical variables.

occupations, lifestyles, housing characteristics, or environmental factors are possible risks for LASV exposure. *M. natalensis* rodents infected with LASV cluster in rural villages with the availability of food between houses. [42, 43, 63, 65] They live communally under floors and walls of houses or in patches of cultivated land. When the rodent population surpasses a density-threshold, there is a higher frequency of human contact, and the risk for LASV exposure increases. [8, 65–67] Houses using poor construction materials allow the population of rodents to increase in density near the household's food and water storage. [8, 65–67] This study found that minimal wall construction (i.e. mud and brick, or mud only used as materials), poor conditions of the household toilet, and floors made only of mud or dirt were significant factors correlating with higher seroprevalence of LASV. *Mastomys* species seasonally migrate from

wild or forested land and cultivated patches in search of food and water. [10, 65, 66] They remain indoors for longer periods as outdoor food sources from wild and cultivated land become scarce. [43, 68] The study found houses in close proximity to bushes, forest areas, water sources, and refuse have higher household seroprevalence, aligning with these patterns.

## A need for age- and sex-based incidence studies

Interactions between rodents and younger people, including hunting and consumption, showed an increased risk of LASV exposure. [11] Previous studies have not determined specific age-related risks regarding general LASV seroprevalence. An odds ratio analysis conducted for age groups of 10 years suggests a 10-year increment in age increased the likelihood an individual will be seropositive; however, this did not persist in multivariate modeling. This could suggest, particularly in endemic areas, consistent prolonged exposure to LASV leads to higher chances of seropositivity as people age. When looking at a breakdown of ages by 10-year age groups, there was a visible trend suggesting this positivity could wane in middle age, though a statistically significant trend was not found. This could be due to multiple reasons, potentially changes in behavior that decrease environmental exposure to LASV or more complicated immunological factors leading to immune senescence in older age. Some LF related studies have suggested differences in ages, however these studies investigated incidence, rather than prevalence. The lack of information about IgG LASV prevalence makes it difficult to make any significant declarations of the correlation between age and seropositivity.

In contrast to previous studies showing no discernible differences between males and females, this study found males were significantly more likely to be seropositive than females across all three districts. [9, 69] This could be related to the transient patterns of men in Sierra Leone or the cultural expectation to embark in occupations related to construction, mining, or transportation. Mining activities encourage migration to mining sites, leading to overcrowded living conditions increasing the risk of infectious diseases; however it is only anecdotally suggested as a risk factor for LASV exposure, so further investigation is necessary. [70] Early studies on the epidemiology of LF shed light on the high prevalence of LASV in regions with extensive diamond mining, but is not entirely understood. [5, 43] This study found miners were slightly more likely to be seropositive than those who worked primarily at home, remaining consistent with previous beliefs, but still requiring more quantified results. This study found a potentially undiscovered increased risk for LASV seroprevalence in transportation-related occupations. The reason for this potential risk is unknown but could be an implication of working similarly to those involved in mining activities. LASV can quickly spread in areas with poor sanitary conditions and overcrowded spaces. [71] Truck drivers and motor-bike drivers travel and sleep away from home often in poorly constructed, potentially rodent-infested areas.

Nosocomial infections are common human-to-human modes of transmission for LF and are important to understand for the safety of healthcare workers. [72] These healthcare workers are generally the first responders to severely ill patients, particularly those with suspected LF. [62] In poorer, under-resourced areas, mitigation practices to prevent transmission may not be available for healthcare facilities. Maintaining proper infection, prevention, and control practices, when possible, can help mitigate this type of transmission. [13] These approaches will also prevent potential outbreaks, which can spread rapidly during the longer incubation period of LASV (7–21 days). [37]

## Limitations

Several limitations should be considered while interpreting the results presented from this study. The case-investigation forms used to determine household and individual

demographics were not previously validated for quantitative and qualitative reliability, resulting in the potential for interviewer or observational bias. This lack of standardization could possibly introduce desirability bias when interpreting the results concerning occupational and environmental factors related to LASV exposure. The large-scale of this study and the length of enrollment could have affected accuracy in reporting, introducing additional bias in the questionnaires. The sample size itself mitigates this issue, but validation should be considered for future studies. Selection of participants relied on the enumeration areas from the 2004 Sierra Leone Census, which were over 10 years old when used for this study. Care should be taken when applying these results and observations to any population of Sierra Leone outside of Kenema, Port Loko, and Tonkolili Districts.

Presence of antibodies suggesting life-time exposure to LASV IgG is shown anecdotally. [48] However, there is documented potential for sero-reversion of LASV IgG seropositivity. [15, 73, 74] As such, this point seroprevalence study could severely underestimate the potential for life-time exposure. At the same time, LASV is known to experience non-specific cross-reactivity between lineages, as well as other related arenaviruses. This could lead to an over-estimation of LASV IgG prevalence in our study population. These issues highlight the need for adequately designed and implemented incidence studies using accurately validated diagnostic kits. There is a gap in surveillance and reporting to the MoHS regarding LASV IgG. We can estimate the incidence of symptomatic LF in the larger population, but this doesn't easily correlate with the seroprevalence of LASV IgG in this population.

The West African Ebola Outbreak occurred immediately before the implementation of this study. Understandably, this outbreak severely affected the Sierra Leone healthcare system, especially the team at KGH, who were intimately involved in the outbreak itself. Despite these limitations, the importance of determining a baseline seroprevalence of LASV in different geographic and administrative areas of Sierra Leone will help future epidemiological, clinical, and public health research.

## Conclusion

Since 2010, LF reported cases have increased (CDC). [17] There are no definitive treatments or vaccines for LF. Vaccine development relies on accurate seroprevalence estimations and strong public health surveillance programs. The heterogeneous distribution of LF and LASV over time and space will make designing efficacy trials and intervention programs difficult. Further understanding of the epidemiology of LASV, specifically in Sierra Leone, is crucial in implementing vaccine trials and public health programs designed to decrease exposure and transmission.

Due to difficulties related to clinical diagnoses, a lack of consistent and adequate surveillance, and rapid, reliable diagnostics, LF and LASV detection remains a challenge. The seroprevalence measured in this study supports the endemicity of LASV in Kenema District and the general perception of risk for LF should be re-evaluated in Tonkolili and Port Loko Districts. The findings indicate LASV IgG seroprevalence is unequally distributed across the three districts studied. The seroprevalence rates within districts varied widely by chiefdom, indicating specific sub-district areas with increased risk of LASV exposure. In addition, this study reinforces the environmental correlates of seroprevalence previously reported. Although this study adds to the literature, the true geographic prevalence and distribution of LASV continues to be difficult to elucidate due to large numbers of asymptomatic cases, non-specific clinical presentations, and human migration and conflict. Identifying geographic hot-spots, or hyper-endemic areas, will greatly assist in efforts to increase awareness and target control programs for LF. This will aid the implementation of future clinical studies and vaccination

development. This study creates a foundational dataset of LASV seroprevalence in Sierra Leone to help guide future studies and interventions.

## Supporting information

**S1 Text. Supplementary tables.** Table A in S1 Text. Selected communities' locations, LASV IgG seroprevalence, and sizes. Table B in S1 Text. List of variables and their descriptions, answer choices, and analysis level collected from inception of the PEER Health study. Table C in S1 Text. LASV IgG Seroprevalence stratified by household, village, and chiefdom level, overall, and between all three Districts. Table D in S1 Text. Univariate analysis of factors associated with potential correlates of exposure to LASV quantified by IgG seropositivity. Table E in S1 Text. Stratified analysis of LASV IgG seropositivity potential correlates of exposure by geographic location, comparing Kenema District and Tonkolili District (reference group). Table F in S1 Text. Stratified analysis of LASV IgG seropositivity potential correlates of exposure by geographic location, comparing Kenema District and Port Loko District (reference group). Table G in S1 Text. Stratified analysis of LASV IgG seropositivity potential correlates of exposure by geographic location, comparing Tonkolili District and Port Loko District (reference group). Table H in S1 Text. Multivariate analysis to determine likelihood of household construction materials and fieldworker observations affecting levels of LASV IgG seroprevalence in a household.
(DOCX)

**S1 Fig. Individual level analysis of housing construction materials and characteristics associated with LASV IgG seropositivity.** (A) Materials used in wall construction; (B) Fieldworker observation of the condition of household toilet facility; (C) Main type of water source used by household; (D) Materials used in floor construction; (E) Materials used in roof construction.
(TIF)

## Acknowledgments

We are incredibly grateful for the hard work of the KGH staff, particularly the Outreach Team and all the individuals who participated in this study. We would like to recognize the support and guidance from the Sierra Leone Ministry of Health and Sanitation and the Sierra Leone Ethics and Scientific Review Committee, without whom this work could not have been done. We would also like to thank members of the Viral Hemorrhagic Fever Consortium (VHFC) and Zalgen Labs for support with diagnostics and immunoassays.

## Author Contributions

**Conceptualization:** Donald S. Grant, Lina Moses, Jeffrey G. Shaffer, Robert F. Garry, John S. Schieffelin.

**Data curation:** Nicole Roberts Yerkes, Foday Alhasan, Adaora Okoli, Ashley Zeoli, Caroline Weldon, Christopher M. Bishop, Jessica Hartnett, Karissa Chao, Kayla Shore, Lilia I. Melnik, Mallory Mucci, Philip Doyle, Rachael Yenni, Rachel Podgorski, Samuel C. Ficenec, Jeffrey G. Shaffer.

**Formal analysis:** Emily J. Engel, Nell G. Bond.

**Funding acquisition:** Donald S. Grant, Lina Moses, Jeffrey G. Shaffer, Robert F. Garry, John S. Schieffelin.

**Investigation:** Donald S. Grant, Emily J. Engel, Lansana Kanneh, Lina Moses, Robert F. Garry, John S. Schieffelin.

**Methodology:** Lansana Kanneh, James Koninga, Michael A. Gbakie, Foday Alhasan, Franklyn B. Kanneh, Ibrahim Mustapha Kanneh, Fatima K. Kamara, Mambu Momoh, Mohamed S. Yillah, Momoh Foday, Jessica Hartnett, Lilia I. Melnik, Rachael Yenni, Lina Moses, Jeffrey G. Shaffer, John S. Schieffelin.

**Project administration:** Emily J. Engel, Lansana Kanneh, James Koninga, Michael A. Gbakie, Foday Alhasan, Franklyn B. Kanneh, Ibrahim Mustapha Kanneh, Fatima K. Kamara, Mambu Momoh, Mohamed S. Yillah, Momoh Foday, Lina Moses, Jeffrey G. Shaffer, John S. Schieffelin.

**Resources:** Lansana Kanneh, James Koninga, Michael A. Gbakie, Foday Alhasan, Franklyn B. Kanneh, Ibrahim Mustapha Kanneh, Fatima K. Kamara, Mambu Momoh, Mohamed S. Yillah, Momoh Foday, Jeffrey G. Shaffer.

**Supervision:** Donald S. Grant, Jeffrey G. Shaffer, John S. Schieffelin.

**Validation:** Emily J. Engel, Nicole Roberts Yerkes, Adaora Okoli, Ashley Zeoli, Caroline Weldon, Christopher M. Bishop, Jessica Hartnett, Karissa Chao, Kayla Shore, Lilia I. Melnik, Mallory Mucci, Nell G. Bond, Philip Doyle, Rachael Yenni, Rachel Podgorski, Samuel C. Ficenec.

**Visualization:** Emily J. Engel, Lina Moses.

**Writing – original draft:** Emily J. Engel.

**Writing – review & editing:** Donald S. Grant, Emily J. Engel, Nicole Roberts Yerkes, Crystal Zheng, Nell G. Bond, Jeffrey G. Shaffer, Robert F. Garry, John S. Schieffelin.

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
