## [Decision Letter · Decision Letter 0]

11 Jul 2022

Dear Ms. Engel,

Thank you very much for submitting your manuscript "Seroprevalence of anti-Lassa Virus IgG antibodies in three districts of Sierra Leone: A cross-sectional, population-based study." for consideration at PLOS Neglected Tropical Diseases. As with all papers reviewed by the journal, your manuscript was reviewed by members of the editorial board and by several independent reviewers. In light of the reviews (below this email), we would like to invite the resubmission of a significantly-revised version that takes into account the reviewers' comments. 

Reviewers 1 and 2 both had some comments on limitations in the data, and how better to acknowledge them or account for them - please modify your discussion appropriately. Reviewer 1 and 2 had some difficulty understanding the statistical models and reporting used - be sure to explain it so that a non specialist reader can understand what was done and why. If any analyses need to be redone after assessing their comments please do so. Were the positive predictive value and sensitivity supposed to be the same number, or was this a typo? Reviewer 3 noted that some of the risk factors are difficult to address because of the retrospective nature of serology, be sure to acknowledge this in your discussion. Reviewer 2 noted that you had not provided precise information on the localities you worked - can this be provided as supplementary data?

We cannot make any decision about publication until we have seen the revised manuscript and your response to the reviewers' comments. Your revised manuscript is also likely to be sent to reviewers for further evaluation.

Sincerely,

Brianna R Beechler, Ph.D., DVM

Academic Editor

Andrea Marzi

Section Editor

Reviewers 1 and 2 both had some comments on limitations in the data, and how better to acknowledge them or account for them - please modify your discussion appropriately. Reviewer 1 and 2 had some difficulty understanding the statistical models and reporting used - be sure to explain it so that a non specialist reader can understand what was done and why. If any analyses need to be redone after assessing their comments please do so. Were the positive predictive value and sensitivity supposed to be the same number, or was this a typo? Reviewer 3 noted that some of the risk factors are difficult to address because of the retrospective nature of serology, be sure to acknowledge this in your discussion. Reviewer 2 noted that you had not provided precise information on the localities you worked - can this be provided as supplementary data?

Reviewer's Responses to Questions

**Key Review Criteria Required for Acceptance?**

**Methods**

-Are the objectives of the study clearly articulated with a clear testable hypothesis stated?

-Is the study design appropriate to address the stated objectives?

-Is the population clearly described and appropriate for the hypothesis being tested?

-Is the sample size sufficient to ensure adequate power to address the hypothesis being tested?

-Were correct statistical analysis used to support conclusions?

-Are there concerns about ethical or regulatory requirements being met?

Reviewer #1: Methods

Line 152-156: this information is already presented in the introduction (see lines 116-124, which is the objective of the study

Line 157: “villages representative of small, medium, and large communities”; please give some clues about this categorization. What is small, the number of inhabitants, the surface? If the demography is the right parameter as I am thinking, so please give a number for each of them. 

Are the small, medium and large communities represented equally in each district? Looking at your figure 1, we can easily estimate the number of individuals per village according to district; Kenema: 172 indiv/village (5162/30), Tonkolili: 118 indiv/village (3198/27) and Port Loko: 91 indiv/village (2278/25). Therefore, it seems that you have a decreasing gradient in sampling your villages from Kenema to Port Loko, showing that the sampling is not uniform, may be due to a higher proportion of large communities in Kenema?

May be due to a lower effort of sampling in the remote places from Kenema?

Line 178-180: the precise location for each household was recorded using a GPS. It is therefore possible to give the coordinates of each village (see my main comment).

Line 188-192: a table gathering all the variables would be very helpful.

Line 201-202: how do you define IgG prevalence? In other words what is the denominator: the individuals? The households?

Line 214: add IgG between “LASV” and “prevalence”

Line 222: you noted that “each replicate required 2-3 mm punches” per individual to run ELISA, but on the field (line 199), you noted that you collected enough blood to fill a single spot. Please harmonize the sampling between the field and the laboratory.

Line 225-226: what is the difference between the sensitivity and positive predictive value, which have the same number? Same question between specificity and negative predictive value? I think this is a redundant information. Also, what is a diagnostic likelihood ratio? What are the comparable commercial diagnostics for Lassa, other pathogens?

Reviewer #2: -Are the objectives of the study clearly articulated with a clear testable hypothesis stated? YES

-Is the study design appropriate to address the stated objectives? YES

-Is the population clearly described and appropriate for the hypothesis being tested? NO, the structure of the source population should be further described.

-Is the sample size sufficient to ensure adequate power to address the hypothesis being tested? YES, probably, but the question of sample size has not been addressed by authors in the manuscript.

-Were correct statistical analysis used to support conclusions? YES

-Are there concerns about ethical or regulatory requirements being met? NO

Reviewer #3: Nigeria, Sierra Leone and Liberia represent the epicentres of LASV epidemics. These regions record most epidemiological studies on LASV. While multiple crucial data on LASV fever are needed (such as factors associated with high case fatality rate (approximately 30%), data on pregnant women, maternal-fetal outcomes, and strong evidence for new treatments and their integration in clinical care), there are already seroprevalence data demonstrating the endemicity of LASV in West Africa including Sierra Leone. Although the study by Grant et al. provides better information on the seroprevalence of LASV in Sierra Leone, it suffers from a lack of novelty and is limited to an extension of knowledge for new regions without being nationally representative of Sierra Leone.

The design of the study, which is also focused in past infections, further weakens the risk factors for acquiring LASV reported in the present study by the probable retrospective nature of the detection of IgG antibodies. Multiple categories of interest for LASV infections such as pregnant women are difficult to consider for these IgG detections because the infection cannot be confirmed as acquired before or after pregnancy.

**Results**

-Does the analysis presented match the analysis plan?

-Are the results clearly and completely presented?

-Are the figures (Tables, Images) of sufficient quality for clarity?

Reviewer #1: Results

Line 238-: give the detailed information about the 82 villages in a table (see my comment above).

Line 261-271: This paragraph is very confusing because several ways of estimating proportions are presented in a descriptive way. The authors begin by using a median value that is not introduced in the M&M and does not reflect Figure 2A (line 261-263). Then they introduce the seroprevalence, which is assumed to be a percentage of the number of IgG positive individuals divided by the number of individuals tested per village. Figures 2B-2D show the percentage of IgG positive households per village, which is not the same. This inflates the percentages and does not allow us to observe the variance in the distribution of IgG.

Line 267-269: What is the point of presenting data by chiefdom if it is not supported by statistics?

Line 270-271: the percentages noted in the text for Tonkolili and Port Loko are different from those in table S1. Furthermore, the numbers of households written in the text are different from those in figure 1; Tonkolili: 272 in figure 1 vs 265 in the text, Port Loko: 198 in figure 1 vs 205 in the text. Please harmonize. 

Line 280-330: it seems that the authors are analyzing the data on positive individuals only

Line 283: the figure 4 is not good because each age class interval is different; 1 year (maybe a few months) for <1 year, 3 years for the 1-4 years interval, 10 years for the 5-14 years interval, 30 years for the 15-44 years interval and unknown for the over 45 years. This figure gives the impression that IgG seroprevalence is very high in the 15-44 age group, but in fact it is disproportionately high compared to the others because it includes the majority of individuals. Equivalent ranges and numbers of individuals per class should be presented in this figure. This would show why the test is not significant (line 283-284). 

Line 320: p=0.013 is noted twice. 

Figures: they are all with a very low resolution. Please provide good ones.

Reviewer #2: -Does the analysis presented match the analysis plan? YES

-Are the results clearly and completely presented? To be improved (cf. comments)

-Are the figures (Tables, Images) of sufficient quality for clarity? NO, unsufficient resolution.

Reviewer #3: (No Response)

**Conclusions**

-Are the conclusions supported by the data presented?

-Are the limitations of analysis clearly described?

-Do the authors discuss how these data can be helpful to advance our understanding of the topic under study?

-Is public health relevance addressed?

Reviewer #1: Discussion

The discussion should include subheadings highlighting the variables of interest. 

I am surprised to see no comparison with the data of McCormick et al. who had already shown a wide distribution of IgG in the country in the 1980s. Indeed, some communities were already infected between 10 and 52% in the eastern province, and between 10 and 15% in the northern province. As your data are not clearly presented, this is of course difficult to compare, especially for the eastern province (= Kenema district). According to your fig 2A, it seems that your IgG seroprevalences are similar to those observed 30 years ago by the McCormick team. This should be said somewhere. 

Line 362-366: this is very speculative. As you have the raw data, this can be verified easily.

Line 427-429: what do you mean by validating a case-investigation form?

Reviewer #2: -Are the conclusions supported by the data presented? NO, not always (cf. comments).

-Are the limitations of analysis clearly described? NO, unsufficiently (cf. comments).

-Do the authors discuss how these data can be helpful to advance our understanding of the topic under study? YES, but in a general way that is quite frustrating.

-Is public health relevance addressed? YES

Reviewer #3: (No Response)

**Editorial and Data Presentation Modifications?**

Reviewer #1: (No Response)

Reviewer #2: Cf. attached comments.

Reviewer #3: (No Response)

**Summary and General Comments**

Reviewer #1: This seroprevalence study of Lassa virus in Sierra Leone was eagerly awaited because the last data were collected in 1980-1985 (Mc Cormick et al. 1987). The authors made a remarkable investigation of 10,642 subjects, distributed in 3 districts. This is twice the sample analyzed by Mc Cormick (n = 5,213) in Sierra Leone and three times the sample analyzed by Lukashevich et al. in 1992 (3,126) in Guinea. Globally, the literature is extensive, accurate and updated.

However, the authors do not mention precisely the localities in which they worked, unlike their predecessors who reported the exact number of people tested per village, while naming the villages. This is one of my major criticism: the authors should present the data in a detailed table including: the district, the chiefdom, the village or locality with their geographical coordinates, the community size and the number of sera positive/tested sera. Indeed, the rough maps showing these locations in Figure 2 are clearly insufficient. Since each household has been geographically located with a Garmin GPS, this should not be a problem. 

The variable "community size" should appear because it is an important epidemiological criterion. Indeed, we can think that in large communities, people are less close to the bushes or fields, which seems to be a risk factor in this study.

There is also a big issue with the data analysis:

- a descriptive analysis taking into account all the results (positive and negative)

- a statistical analysis (multivariate) taking into account the positive results only. 

Why doesn't the multivariate analysis take into account all the data (positive + negative)? This would allow for a robust analysis by taking into account the variance of each parameter. Because presented this way, one wonders why the variables are analyzed several times? For example, the "district" variable was analyzed twice: line 260-271, then line 289-303. 

It also seems that the multivariate analysis in the results (line 280-307) presents the data by individual, while the second part (line 308-330) presents the data by household. At the end, we don't know which parameters are relevant and which are really significant because of the change of outcome variable between one analysis and the other!

Abstract

Line 30: the keywords do not describe well your study. Pease add some others, such as IgG, prevalence, housing quality, etc

Introduction

Line 83: “the multimammate rat” should be “the Natal multimammate mouse”

Line 89: why do you write LF and LASV infection rates? I think this is LASV infection only.

Line 90-92: please update the numbers with more recent studies.

Line 124-130: these 4 sentences present the main findings of the study, as it is fine in a summary. This is inaccurate at the end of the introduction. Please change this part.

Reviewer #2: Cf. attached comments.

Reviewer #3: (No Response)

PLOS authors have the option to publish the peer review history of their article (what does this mean?). If published, this will include your full peer review and any attached files.

Reviewer #1: No

Reviewer #2: No

Reviewer #3: No
---

## [Decision Letter · Decision Letter 1]

19 Oct 2022

Dear Ms. Engel,

Thank you very much for submitting your manuscript "Seroprevalence of anti-Lassa Virus IgG antibodies in three districts of Sierra Leone: A cross-sectional, population-based study." for consideration at PLOS Neglected Tropical Diseases. As with all papers reviewed by the journal, your manuscript was reviewed by members of the editorial board and by several independent reviewers. The reviewers appreciated the attention to an important topic. Based on the reviews, we are likely to accept this manuscript for publication, providing that you modify the manuscript according to the review recommendations. 

One reviewer recommends some editorial changes and clarifications that will improve the manuscript - including careful wording of lassa fever vs. virus and some statistical method clarifications. As editor, I find figure 5 easy enough to understand, but if you have any clarifications that you think will help the reviewer who did not like it please add them. Otherwise you can disregard. One of the other reviewers suggests several papers you could cite in your literature review, if you wish to you can, but it is not required.

Sincerely,

Brianna R Beechler, Ph.D., DVM

Academic Editor

Andrea Marzi

Section Editor

One reviewer recommends some editorial changes and clarifications that will improve the manuscript - including careful wording of lassa fever vs. virus and some statistical method clarifications. As editor, I find figure 5 easy enough to understand, but if you have any clarifications that you think will help the reviewer who did not like it please add them. Otherwise you can disregard. One of the other reviewers suggests several papers you could cite in your literature review, if you wish to you can, but it is not required.

Reviewer's Responses to Questions

**Key Review Criteria Required for Acceptance?**

**Methods**

-Are the objectives of the study clearly articulated with a clear testable hypothesis stated?

-Is the study design appropriate to address the stated objectives?

-Is the population clearly described and appropriate for the hypothesis being tested?

-Is the sample size sufficient to ensure adequate power to address the hypothesis being tested?

-Were correct statistical analysis used to support conclusions?

-Are there concerns about ethical or regulatory requirements being met?

Reviewer #1: (No Response)

Reviewer #2: -Are the objectives of the study clearly articulated with a clear testable hypothesis stated? YES

-Is the study design appropriate to address the stated objectives? YES

-Is the population clearly described and appropriate for the hypothesis being tested? YES

-Is the sample size sufficient to ensure adequate power to address the hypothesis being tested? UNCERTAIN

-Were correct statistical analysis used to support conclusions? YES

-Are there concerns about ethical or regulatory requirements being met? NO

Reviewer #3: (No Response)

**Results**

-Does the analysis presented match the analysis plan?

-Are the results clearly and completely presented?

-Are the figures (Tables, Images) of sufficient quality for clarity?

Reviewer #1: (No Response)

Reviewer #2: -Does the analysis presented match the analysis plan? YES

-Are the results clearly and completely presented? NO, clarity should be improved

-Are the figures (Tables, Images) of sufficient quality for clarity? NO, maps are poorly readable

Reviewer #3: (No Response)

**Conclusions**

-Are the conclusions supported by the data presented?

-Are the limitations of analysis clearly described?

-Do the authors discuss how these data can be helpful to advance our understanding of the topic under study?

-Is public health relevance addressed?

Reviewer #1: (No Response)

Reviewer #2: -Are the conclusions supported by the data presented? YES

-Are the limitations of analysis clearly described? YES, but some aspects still need to be discussed (see attached comments)

-Do the authors discuss how these data can be helpful to advance our understanding of the topic under study? YES

-Is public health relevance addressed? YES

Reviewer #3: (No Response)

**Editorial and Data Presentation Modifications?**

Reviewer #1: (No Response)

Reviewer #2: Minor comments

P9, line 41 please correct “Lassa fever virus”=> “Lassa virus”.

P11, line 91. I would say “estimated” rather than “approximately” as these assumptions are based on very ancient epidemiological studies conducted in the 1980s. 

P12, line 116. It is not necessary to say “antibodies” as this term is implicit in “seroprevalence”. 

P15, line 195. Please add Fullah to the list of languages, as indicated P13, line 141.

P20, line 296. Who is “that individual”?

Reviewer #3: (No Response)

**Summary and General Comments**

Reviewer #1: The authors did a major revision, incorporating all my comments. 

I have one last request regarding the figure 4; can you add the numbers of people per age class above each column.

Reviewer #2: Major comments

P12, lines 117-118. The terms “identify correlates of prevalence for LASV acquisition” are unclear. Please clarify. Same for “potential correlates of prevalence related to LASV acquisition”, P13 line 147.

P12, line 118. Please add “nucleoprotein” between “Anti-LASV” and “IgG ELISAs” as previously requested.

P14, Table 1. There is a confusion between the prevalence of “Lassa fever”, being the acute symptomatic disease due to the infection with LASV, and the seroprevalence of “LASV IgG”. Please correct throughout the table and elsewhere in the manuscript. Also, please clarify “15 percentage points with 95% confidence”. Lastly, please clarify if “10 or 20 households” is the number of investigated households per selected villages; as it stands one could understand that it is the absolute number of investigated households.

P16, line 216. “Seroprevalence/seropositivity” are the “dependent” and not the “independent” variables. Moreover, interactions terms are usually set when there is a suspicion of an interaction between two “dependent”/”explanatory” variables, and not between two “independent”/”explained” variables. Please correct and clarify.

P16, line 218 and thereafter. Please indicate from the beginning of this section the detected LASV IgG epitope. I guess “nucleoprotein” in the present case. This is a key information for readers in terms of generalizability and reproducibility.

Results section (in general). It is still not clear in the way the results are presented what apply to households (seroprevalence) and to individuals (seropositivity). The results should be presented in such a way as to avoid any confusion in the mind of the reader and to take up the distinction between households and individuals introduced in the methods.

P19, line 268 and thereafter. The sentence “Kenema District had 10 villages with greater than 20% seroprevalence, which was higher than both Port Loko and Tonkolili Districts (33.0% (n=30), Fig 2B).” is disturbing. Please homogenize and present both absolute valued and percentage for each district rather than giving absolute values for one and percentage for the others. The same apply to the two sentences that follow.

P20, lines 298 to 300. The authors state “Though not statistically significant, individuals in Tonkolili District were less likely to test positive for LASV IgG than those in Port Loko District (OR 0.812, p=.360, S4 Table).”

If not significant (and it is clearly not), it is not correct to imply that there is a difference in seropositivity between individuals living in the two districts. Please remove.

Same comment for the sentence P20, line 316-317.

P23, lines 362 to 364. The authors compare their results to those of the McCormick’s study conducted in the 1980s. However, the results of both studies are not directly comparable, notably because the methods used to determine seropositivity (IFA in the McCormick’s study and ELISA detecting IgG against a specific epitope in the present study) are very different and would probably have very different diagnostic performances if applied in parallel to the samples collected here. This should be clearly explained in the discussion.

Figure 5. This figure is very difficult to read and understand. Please consider another way of presenting those results.

Reviewer #3: The authors did address the limitations raised except for claiming that the latest studies on the epidemiology of LASV in Sierra Leone date from the 1980s, notably the studies conducted by McCormick et al. There are studies with recent data such as those cited below that I would suggest authors consider in their discussion;

O’Hearn, A.E., Voorhees, M.A., Fetterer, D.P., Wauquier, N., Coomber, M.R., Bangura, J., Fair, J.N., Gonzalez, J.-P., Schoepp, R.J., 2016. Serosurveillance of viral pathogens circulating in West Africa. Virol J 13, 163. https://doi.org/10.1186/s12985-016-0621-4.

Schoepp, R.J., Rossi, C.A., Khan, S.H., Goba, A., Fair, J.N., 2014. Undiagnosed acute viral febrile illnesses, Sierra Leone. Emerg Infect Dis 20, 1176–1182. https://doi.org/10.3201/eid2007.131265.

PLOS authors have the option to publish the peer review history of their article (what does this mean?). If published, this will include your full peer review and any attached files.

Reviewer #1: No

Reviewer #2: No

Reviewer #3: No

Figure Files:

Data Requirements:

Reproducibility:

References

---

## [Editor Report · Decision Letter 2]

9 Nov 2022

Dear Ms. Engel,

We are pleased to inform you that your manuscript 'Seroprevalence of anti-Lassa Virus IgG antibodies in three districts of Sierra Leone: A cross-sectional, population-based study.' has been provisionally accepted for publication in PLOS Neglected Tropical Diseases.

Best regards,

Brianna R Beechler, Ph.D., DVM

Academic Editor

Andrea Marzi

Section Editor

---

## [Editor Report · Acceptance letter]

17 Jan 2023

Dear Ms. Engel,

We are delighted to inform you that your manuscript, "Seroprevalence of anti-Lassa Virus IgG antibodies in three districts of Sierra Leone: A cross-sectional, population-based study.," has been formally accepted for publication in PLOS Neglected Tropical Diseases.

Best regards,

Shaden Kamhawi

co-Editor-in-Chief

Paul Brindley

co-Editor-in-Chief
